# Different Cell Responses to Hinokitiol Treatment Result in Senescence or Apoptosis in Human Osteosarcoma Cell Lines

**DOI:** 10.3390/ijms23031632

**Published:** 2022-01-31

**Authors:** Shun-Cheng Yang, Hsuan-Ying Chen, Wan-Ling Chuang, Hui-Chun Wang, Cheng-Pu Hsieh, Yi-Fu Huang

**Affiliations:** 1Division of Pediatric Infection, Changhua Christian Children’s Hospital, Changhua 50006, Taiwan; 61359@cch.org.tw; 2Orthopedics & Sports Medicine Laboratory, Changhua Christian Hospital, Changhua 50006, Taiwan; 180342@cch.org.tw; 3Surgery Clinical Research Center, Changhua Christian Hospital, Changhua 50006, Taiwan; 159260@cch.org.tw; 4Graduate Institute of Natural Products, College of Pharmacy, Kaohsiung Medical University, Kaohsiung 80708, Taiwan; wanghc@kmu.edu.tw; 5Department of Orthopedic Surgery, Changhua Christian Hospital, Changhua 50006, Taiwan; 6Department of Post-Baccalaureate Medicine, College of Medicine, National Chung Hsing University, Taichung 40227, Taiwan; 7Department of Kinesiology Health Leisure Studies, Chienkuo Technology University, Changhua 50094, Taiwan

**Keywords:** hinokitiol, senescence, apoptosis, osteosarcoma

## Abstract

Hinokitiol is a tropolone-related compound isolated from the heartwood of cupressaceous plants. It is known to exhibit various biological functions including antibacterial, antifungal, and antioxidant activities. In the study, we investigated the antitumor activities of hinokitiol against human osteosarcoma cells. The results revealed that hinokitiol treatment inhibited cell viability of human osteosarcoma U-2 OS and MG-63 cells in the MTT assay. Further study revealed that hinokitiol exposure caused cell cycle arrest at the S phase and a DNA damage response with the induction of γ-H2AX foci in both osteosarcoma cell lines. In U-2 OS cells with wild-type tumor suppressor p53, we found that hinokitiol exposure induced p53 expression and cellular senescence, and knockdown of p53 suppressed the senescence. However, in MG-63 cells with mutated p53, a high percentage of cells underwent apoptosis with cleaved-PARP expression and Annexin V staining after hinokitiol treatment. In addition, up-regulated autophagy was observed both in hinokitiol-exposed U-2 OS and MG-63 cells. As the autophagy was suppressed through the autophagy inhibitor chloroquine, hinokitiol-induced senescence in U-2 OS cells was significantly enhanced accompanying more abundant p53 expression. In MG-63 cells, co-treatment of chloroquine increased hinokitiol-induced apoptosis and decreased cell viability of the treated cells. Our data revealed that hinokitiol treatment could result in different cell responses, senescence or apoptosis in osteosarcoma cell lines, and suppression of autophagy could promote these effects. We hypothesize that the analysis of p53 status and co-administration of autophagy inhibitors might provide more precise and efficacious therapies in hinokitiol-related trials for treating osteosarcoma.

## 1. Introduction

Osteosarcoma is the most common type of primary malignant bone tumor, mainly identifying in adolescents and young adults. Surgical resection is the major treatment for osteosarcoma, but survival rate of the patients treated with surgery alone is lower than 20%. In the 1970s, effective chemotherapeutic agents, including adriamycin, cisplatin, epirubicin, and methotrexate, were applied as adjuvant treatments to facilitate surgical resection. This integration has improved overall 5-year survival rates that approach 70%. Unfortunately, there has been no further progress in this outcome over the past decades. Therefore, identification of effective drugs and strategies are urgently required for patients with osteosarcoma [1,2].

Hinokitiol is a natural monoterpenoid isolated from the heartwood of cupressaceous plants, and is well-known for its antimicrobial abilities [3,4]. In addition, the antitumor activity of hinokitiol has been shown in some types of cancer cells. The literature shows that hinokitiol suppresses cell viability and decreases survivin expression by the ERK/MKP3/proteosome pathway in B16 melanoma [5]. Hinokitiol inhibits migration and reduces MMP-2 and MMP-9 activities in A549 lung cancer cells [6]. Hinokitiol up-regulates the level of reactive oxygen species and induces apoptosis in endometrial cancer cell lines [7]. Hinokitiol represses the growth of hepatocellular carcinoma through autophagic cell death, apoptosis, and cell cycle arrest [8]. Hinokitiol inhibits the growth of basal-like mammary tumors by regulating GSK-3β/β-catenin signaling [9]. Hinokitiol decreases the expression of heparanase through the Erk and Akt pathways to suppress tumor metastasis in mouse melanoma B16F10 cells and breast cancer 4T1 cells [10]. Hinokitiol reduces cell growth in human colon cancer cells by triggering S-phase arrest and apoptosis [11]. Hinokitiol significantly eliminates the hallmarks of glioma stem cells via the downregulation of Nrf2 expression [12].

Cellular senescence is a highly heterogeneous state characterized as irreversible growth arrest, flattened and enlarged morphologies, resistance to apoptotic stimuli, deregulated metabolism, and a complex proinflammatory secretory phenotype. In addition, the activity of senescence-associated β-galactosidase, a lysosomal enzyme, is increased in senescent cells. Senescence can be induced by DNA damage response, mitochondrial dysfunctionality or oxidative stress [13,14]. Cellular senescence can be modulated primarily through the p53/p21 and p16/RB pathways [14].

Transcription factor p53 is a crucial tumor suppressor that is mutated in more than 50% of all tumors, including osteosarcoma. It can be activated through various stresses, including DNA damage, oncogene expression, hypoxia, and replication stress, as well as cellular metabolic changes. Once activated, p53 can trigger apoptosis, cell cycle arrest, or senescence to suppress tumor progression by activating its target genes [15,16]. *CDKN1A* is the critical induced gene in p53-mediated cellular senescence. This gene codes for the p21^WAF1^ protein, a cyclin-dependent kinase (CDK) inhibitor. p21 interacts and inhibits CDK complexes directly, and thus suppresses the phosphorylation of Rb and activation of E2F, a transcription factor driving cell-cycle progression, resulting in growth arrest [17].

Different serine, threonine, and tyrosine phosphorylation sites have been identified on the N-terminal domains of p53. The phosphorylation of these sites, including serine15 and tyrosine18, could increase p53 stability by promoting its dissociation from the MDM2 ubiquitin ligase [18,19]. Several protein kinases have shown the ability to phosphorylate p53 directly. For example, ATM (ataxia-telangiectasia mutated) or ATR (ATM- and Rad3-Related) phosphorylate p53 at serine 15 in response to DNA damage [19]. TTK/hMps1 phosphorylates p53 at Thr18 after spindle damage [18].

Several research groups have tried to demonstrate the correlation between p53 status and prognosis in human osteosarcoma. Nevertheless, the outcome is still controversial. Chen, et al. showed that the patients with mutated p53 may have a poor 2-year overall survival rate compared to the patients with wild type p53 [20]. A different group conducted a meta-analysis in which expression of p53 protein was correlated with an unfavorable prognosis of patients with osteosarcoma in overall survival [21]. In this report, we examined the anticancer activity of hinokitiol against osteosarcoma using U-2 OS cells (wild type p53) and MG-63 cells (mutated p53).

## 2. Results

### 2.1. Hinokitiol Reduces Viability and Proliferation of Osteosarcoma Cell Lines

The cytotoxic effect of hinokitiol on human osteosarcoma U-2 OS and MG-63 cells was determined by MTT assay. The data showed that the treatment with hinokitiol decreased the viability of U-2 OS or MG-63 cells at concentration of 10 to 80 μM after treatment for 48 h, and U-2 OS cells were more sensitive to this drug (Figure 1A). The half maximal inhibitory concentration (IC50) for U-2 OS cells were 44 μM at 24 h and 25 μM at 48 h, and those for MG-63 cells were 36 μM at 48 h. Furthermore, a colony formation assay was performed to demonstrate the influence of hinokitiol on the proliferative potential of a single cell. The data showed that hinokitiol exposure reduced colony formation ability in a dose-dependent manner (Figure 1B). Overall, the data indicate that hinokitiol exposure decreases viability and proliferation of osteosarcoma cell lines. A dosage range of 20 to 80 μM was chosen for the current study. These doses were less cytotoxic in human lung fibroblast MRC-5 cells (Figure 1C) and other normal cells [22]. The IC50 for MRC-5 was 72 μM at 48 h.

### 2.2. Hinokitiol Induces S-Phase Arrest of Cell Cycle and DNA Damage Response

To demonstrate whether an antiproliferative effect of hinokitiol in osteosarcoma cell lines was associated with abnormal cell-cycle progression, cell-cycle analysis was performed by flow cytometry. The data showed that the proportion of cells in the S phase was significantly increased after hinokitiol exposure. At the same time, the proportion of cells in the G1 phase was decreased compared with untreated group. The results indicate that hinokitiol exposure resulted in an increased proportion of S-phase cells in U-2 OS and MG-63 cells (Figure 2A,B). Furthermore, we performed an Edu incorporation experiment to verify the effect of hinokitiol on cell-cycle arrest. As shown in Figure 2C, the newly incorporated Edu-labeled cells were largely down-regulated in hinokitiol-exposed cells (Figure 2C). Overall, our data indicate that hinokitiol exposure induces a cell-cycle arrest at S phase in U-2 OS and MG-63 cells.

A DNA damage response has been shown to have an effect on inhibition of cell proliferation and S-phase arrest of the cell cycle [18]. To assess the mechanism involved in hinokitiol-induced S phase arrest in U-2 OS and MG-63 cell lines, we investigated the expression of DNA damage response-related proteins by Western blotting. The data revealed that hinokitiol exposure resulted in rapid phosphorylation of ATM (at serine 1981), ATR (at ser 428), Chk1 (at serine 345), Chk2 (at threonine 68), and γ-H2AX (Figure 3A). In addition, we found that γ-H2AX formed distinct nuclear foci after hinokitiol treatment in both cell lines (Figure 3B). These data suggest that the DNA damage response may contribute to hinokitiol-induced S phase arrest of the cell cycle in U-2 OS and MG-63 cells.

### 2.3. Hinokitiol Induces Senescence in U-2 OS Cells

As U-2 OS cells were exposed to hinokitiol, we observed a flattened and enlarged morphologies in the treated cells. These characteristics were similar to the senescence phenotype. To evaluate the effect of hinokitiol exposure on senescence, the staining of senescence-associated β-galactosidase (SA-β-gal) was performed in treated U-2 OS cells. The data showed that the treated-U-2 OS cells were strongly stained blue compared with untreated cells, suggesting that hinokitiol treatment could trigger cellular senescence in U-2 OS cells (Figure 4A).

Furthermore, we found that the tumor suppressor p53 and its downstream gene, p21, could be activated in hinokitiol-exposed U-2 OS cells (Figure 4B). To examine the roles of p53 in hinokitiol-induced senescence, the stable Tet-on U2OS-shp53 line [15], which possesses a Tet operator-driven short hairpin RNA (shRNA) that targets p53, was applied. The results showed, in the shp53-expressing cells (with tetracycline), that the protein level of p53 was efficiently attenuated in hinokitiol-treated U-2 OS cells. Once p53 was down-regulated, the level of p21 and the senescence triggered by hinokitiol exposure were significantly decreased (Figure 4A,B). In addition, the level of the senescence marker, plasminogen activator inhibitor-1 (PAI-1) [23], was increased in hinokitiol-treated U-2 OS cells as well, and may be regulated by p53 in these cells (Figure 4B).

We also assessed the influence of hinokitiol on apoptosis in U-2 OS cells by staining with Annexin V and propidium iodide (PI). The data shows that there were a few increases in Annexin V-positive cells as U-2 OS cells were incubated with hinokitiol (40 or 80 μm) for 48 h (Figure 4C). Overall, the results indicate that hinokitiol exposure triggers a p53-dependent senescence in U-2 OS cells.

### 2.4. Hinokitiol Triggers Apoptosis in MG-63 Cells

As MG-63 cells were exposed to hinokitiol, we also observed a flattened morphology in the treated cells, with some round cells in higher dose (80 μm) (Figure 5A). To analyze whether there was the same senescent response that occurred in U-2 OS cells, the staining of SA-β-gal was performed in treated MG-63 cells. As shown in Figure 5A, the stained blue color in cells was weaker, and the percentage of the stained cells was not greatly increased compared to untreated cells (Figure 5A). Based on this finding, we tested the effect of hinokitiol on apoptosis in MG-63 cells. The data showed that hinokitiol exposure induced expression of cleaved PARP and caspase 3, and effectively increased the ratio of Annexin V-positive cells in treated MG-63 cells, indicating hinokitiol exposure up-regulates apoptosis in MG-63 cells.

### 2.5. The Suppression of Autophagy Enhances Hinokitiol-Induced Response

We examined whether autophagy was involved in hinokitiol-induced response in osteosarcoma cell lines. The results revealed that hinokitiol exposure up-regulated expression of LC3-II and enhanced LC3 puncta formation in U-2 OS and MG-63 cells (Figure 6A,B and Figure 7A,B), indicating autophagy is induced in both cell lines after treatment. Furthermore, as autophagy was suppressed by chloroquine in U-2 OS cells, SA-β-gal positive cells (Figure 6C,D) and p53 expression (Figure 6E) were increased, indicating suppression of autophagy enhances p53-dependent senescence in hinokitiol-exposed U-2 OS cells. In addition, co-treatment of chloroquine in MG-63 cells increased the levels of cleaved PARP and caspase 3 (Figure 7C), and decreased cell viability (Figure 7D), indicating that attenuation of autophagy up-regulates apoptosis in hinokitiol-exposed MG-63 cells.

## 3. Discussion

Cellular senescence could be induced primarily through p53 and p16 pathways. The literature has shown that hinokitiol treatment induces senescence in lung adenocarcinoma H1975 cells. Although induced γ-H2AX was observed after hinokitiol treatment for 24 or 48 h, p53 was not activated in H1975 cells that possessed a wild type p53 [24]. In our report, hinokitiol exposure resulted in rapid phosphorylation of ATM (at serine 1981), ATR (at ser 428), Chk1 (at serine 345), Chk2 (at threonine 68), and γ-H2AX, and induced expressions of p53 and p21 in U-2 OS cells. Knockdown of p53 could suppress hinokitiol-induced senescence in U-2 OS cells. It was suggested that the mechanisms involved in regulating hinokitiol-induced senescence may be different between the two cell lines.

DNA damage response (DDR) starts with the recognition of DNA lesions by sensor proteins. This action recruits and activates the most upstream DDR kinases, and thus turns on a cellular signaling cascade to arrest the cell cycle, activate DNA repair, and to remove cells with unrepairable genomes. ATM (ataxia-telangiectasia mutated), ATR (ATM- and Rad3-Related), and DNA-PKcs (DNA-dependent protein kinase) kinases are the most upstream DDR kinases. ATM and DNA-PKcs are activated by double-stranded DNA breaks, whereas ATR is activated by a broad spectrum of genotoxic stresses that induces single-stranded DNA (ssDNA) [25,26]. Our data indicated that hinokitiol exposure resulted in rapid phosphorylation of ATM and ATR. However, it was uncertain how hinokitiol induced DDR, and what kinds of DNA lesions were induced.

Autophagy is a lysosome-dependent self-digestive program. It deals with damaged or useless proteins, organelles, or other cytoplasmic components to restore energy balance. The role of autophagy in regulating cancer cell death or survival remains divergent, and may depend on cellular contents [27]. The literature shows that suppression of autophagy inhibits hinokitiol-induced senescence in lung adenocarcinoma H1975 cells [24]. Suppression of autophagy increased cell viability that inhibited by hinokitiol treatment in murine breast and colorectal cancer cell lines [28] and human hepatocellular carcinoma cell lines [8]. In this study, we found that autophagy was activated as a protective mechanism to mediate hinokitiol-induced responses in osteosarcoma cell lines. Suppression of autophagy up-regulated hinokitiol-induced senescence in U-2 OS cells, and enhanced apoptosis in hinokitiol-treated MG-63 cells.

The initial demonstration of cell senescence considers senescence as an important mechanism of tumor suppression. Senescence could arrest the proliferation of cancer cells and facilitate the clearance of affected cells through immunosurveillance. Recent studies show most senescent cells secrete diverse inflammatory cytokines, chemokines and proteases, which are referred to as the senescent associated secretory phenotype (SASP). The SASP factors could either stimulate or suppress tumor growth and progression depending on their exact compositions. The compositions are determined by different cell types and drug treatment, and may change with time [13,29]. Our data showed that hinokitiol exposure inhibited growth of osteosarcoma cell lines through senescence or apoptosis. However, the limitation of this study is that all results were from in vitro experiments. Recent studies have shown anticancer activity of hinokitiol against different tumors in a mouse model. In these studies, the effective dose was ranged from 2 mg to 100 mg/kg/day in mice, and the data shows that hinokitiol is relatively nontoxic to the animals even at higher doses [8,11,24]. Whether hinokitiol could inhibit the growth of osteosarcoma and what is effective dose in in vivo remains to be demonstrated. In addition, that literature has shown anti-inflammatory properties of hinokitiol in MG-63 cells [30]. Inflammation plays a critical role in tumor progression, and anti-inflammation has been considered a good strategy for cancer therapy [31]. Whether this property is involved in anticancer activity of hinokitiol against osteosarcoma should be further addressed in an in vivo study.

In conclusion, our data showed that hinokitiol exposure suppressed cell viability, and induced S-phase arrest of the cell cycle and the DNA damage response in osteosarcoma cell lines. p53-dependent senescence was observed in osteosarcoma U-2 OS cells, and apoptotic cells were found in osteosarcoma MG-63 cells that possess mutated p53 in response to hinokitiol treatment. Suppression of autophagy largely enhanced senescence in U-2 OS cells and apoptosis in MG-63 cells (Figure 8). We hypothesize that the analysis of p53 status and co-administration of autophagy inhibitors may produce more specific and effective therapies in hinokitiol-related trials for treating osteosarcoma.

## 4. Materials and Methods

### 4.1. Cell Culture and Reagents

Human osteosarcoma U-2 OS, MG-63 cells, and human lung fibroblast MRC-5 cells were purchased from the Bioresource Collection and Research Center (Hsinchu, Taiwan). U-2 OS cells were cultured in McCoy’s 5A medium, and MG-63 cells and MRC-5 were maintained in Minimum Essential Medium supplemented with 10% fetal bovine serum, 100 U/mL penicillin/streptomycin, and maintained at 37 °C in a humidified atmosphere of 5% CO_2_. The passage 97th to 112nd of MG-63 cells, and the passage 28th to 33rd of MCR-5 were used in this study. The stable Tet-on U-2 OS-shp53 cell line was generated as previously described [18]. Hinokitiol were purchased from Cayman Chemical (Ann Arbor, MI, USA), and the purity was greater than 98%.

### 4.2. Colony Formation Assay

U-2 OS cells (2 × 10^5^ cells/dish) were seeded in 35 mm dish for overnight and then incubated with different doses of hinokitiol for 6 h. The treated cells were washed by PBS, trypsinized, and then five thousand cells were cultured onto 35 mm dishes with drug-free complete medium for 10 days to allow colony formation. Colonies were stained by 1% crystal violet solution before counting.

### 4.3. Cell Viability Assay

The effect of Hinokitiol on cell viability was evaluated using MTT (3-(4,5-dimethylthiazol-2-y1)-2,5-diphenyltetrazolium bromide). U-2 OS (4 × 10^4^ cells/well) and MG-63 cells (3 × 10^4^ cells/well) were cultured in 24-well plates for overnight and then incubated with different doses of hinokitiol for the indicated times The details are described in previous report [32].

### 4.4. Flow Cytometry Analysis

U-2 OS (2 × 10^5^ cells/dish) and MG-63 cells (1.5 × 10^5^ cells/dish) were seeded in 35 mm dish for overnight and then exposed to different doses of hinokitiol for 24, 48, or 72 h. The treated cells were collected to analyze DNA content. The details are described in previous report [32].

### 4.5. Apoptosis Assay

U-2 OS (2 × 10^5^ cells/well) and MG-63 cells (1.5 × 10^5^ cells/well) were seeded in 35 mm dish for overnight, and then exposed to different doses of hinokitiol for 48 h. Cell apoptosis was determined by flow cytometry using the Annexin-V-FITC staining kit (Becton Dickinson, San Jose, CA, USA) according to the manufacturer’s instructions as described in a previous report [32].

### 4.6. Cell Lysis and Immunoblotting

The details are described in previous report [32]. For Western blotting, the used antibodies were as following: p21 (OP79; Oncogene Science), p-ATM (Ser1981) (sc-47739; Santa Cruz, CA, USA), p53 (sc-98, Santa Cruz), cleaved caspase-3 (#9661; Cell Signaling, Danvers, MA, USA), phospho-Chk2 (Thr68) (#2197; Cell Signaling), Chk2 (#05-649; EMD Millipore, Temecula, CA, USA). LC3A/B (#12741; Cell Signaling), Chk1 (#2345; Cell Signaling), phospho-Chk1 (Ser345) (#2348; Cell Signaling), phospho-Histone H2A.X (Ser139) (#9718; Cell Signaling), ATM (GTX70103; GeneTex, Irvine, CA, USA), GAPDH (#2118, Cell Signaling), PARP (#9542; Cell Signaling), phospho-p53 (Ser15) (#9284; Cell Signaling). phospho-ATR (Ser428) (#2853; Cell Signaling). PAI-1 (#612024; BD Transduction Laboratories, Lexington, KY, USA).

### 4.7. EdU Incorporation Assay

Edu (5-ethynyl-20-deoxyuridine) was applied to detect newly DNA synthesis. The cells were seeded on glass coverslips with poly-l-lysine coating. The cells were incubated with hinokitiol (40 μM) for 48 h, and then incubated with EdU (10 μM) for 30 min. The cells were harvested and analyzed by Click-iT EdU imaging kit (Invitrog en, Carlsbad, CA, USA) according to the manufacturer’s protocol.

### 4.8. Immunofluorescence Staining

To detect γ-H2AX foci or LC3 puncta formation, the treated cells were immunostained as previously described [33].

### 4.9. Senescence-Associated Beta-Galactosidase (SA-β-gal) Staining

U-2 OS (2 × 10^5^ cells/well) and MG-63 cells (1.5 × 10^5^ cells/well) were seeded in a35 mm dish overnight. The cells were exposed to hinokitiol (40 μM) for 48 h, and senescent cells were determined using the Senescence β-Galactosidase Staining Kit (#9860, Cell Signaling) according to the manufacturer’s protocol.

### 4.10. Statistical Analysis

All experiments were repeated at least three times independently. Statistical significance was assessed by GraphPad Prism 4 (GraphPad Software; San Diego, CA, USA) using the Student’s *t*-test. *p* values of 0.05 were considered significant.

## Figures and Tables

**Figure 1 ijms-23-01632-f001:**
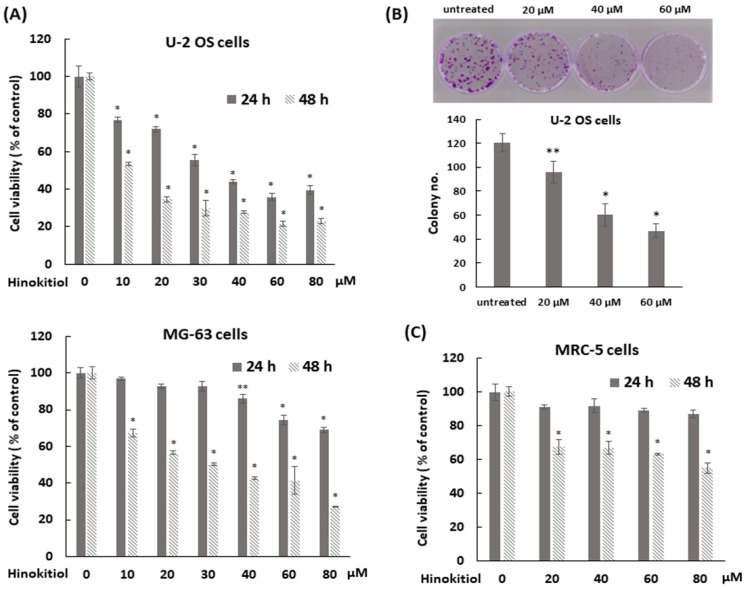
Hinokitiol suppresses cell viability of human osteosarcoma cell lines. (**A**) Hinokitiol inhibits cell growth of osteosarcoma U-2 OS and MG-63 cell lines. U-2 OS or MG-63 cells were incubated with different dose of hinokitiol for 24 or 48 h. MTT assays were performed to measure cell viability. (**B**) Hinokitiol inhibits colony formation of osteosarcoma cell line. After administration of hinokitiol for 6 h, U-2 OS cells were washed by PBS, trypsinized, and then five thousand cells were cultured onto 35 mm dishes with drug-free complete medium for 10 days to allow colony formation. Colonies were stained by 1% crystal violet solution before counting. Quantitative data of colony number/dish were estimated as below. Data are expressed as mean ± SD of three independent experiments. (**C**) The effect of hinokitiol on human lung fibroblast MRC-5 cells. MRC-5 cells were incubated with different doses of hinokitiol for 24 or 48 h. MTT assays were performed to measure cell viability. * *p* < 0.001, and ** *p* = 0.001.

**Figure 2 ijms-23-01632-f002:**
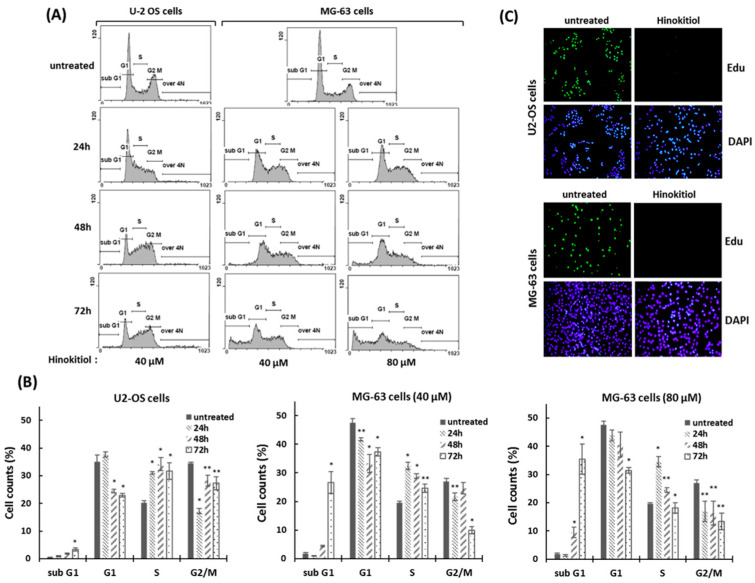
Hinokitiol induces S phase arrest in osteosarcoma cell lines. (**A**) Hinokitiol treatment results in accumulation of osteosarcoma cells in the S phase of the cell cycle. U-2 OS and MG-63 were incubated with hinokitiol for 24, 48, or 72 h. DNA contents of cells were determined by propidium iodide (PI) staining using flow cytometry. Quantitative results of different cell populations were revealed (**B**). * *p* < 0.001, and ** *p* < 0.01. (**C**) Hinokitiol suppresses proliferation of osteosarcoma cell lines. U-2 OS and MG-63 cells were incubated with hinokitiol (40 μm) for 24 h. New DNA synthesis in these cells was evaluated through the Edu incorporation assay.

**Figure 3 ijms-23-01632-f003:**
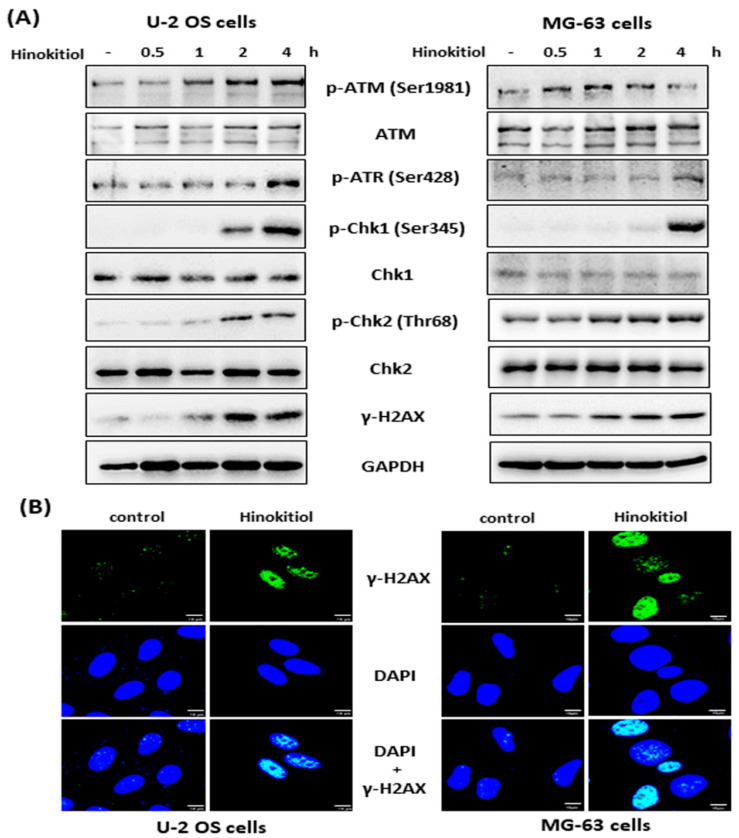
Hinokitiol induces a DNA damage response in osteosarcoma cell lines. U-2 OS or MG-63 cells were incubated with hinokitiol (40 μM), and collected at the indicated time points. The phosphorylation level of ATM, ATR, Chk1, Chk2, and γ-H2AX were analyzed by Western blotting (**A**). (**B**) Hinokitiol induces γ-H2AX foci. U-2 OS or MG-63 cells were incubated with hinokitiol (40 μM) for 4 h, and then the cells were immunostained for γ-H2AX and nuclei were counterstained with DAPI.

**Figure 4 ijms-23-01632-f004:**
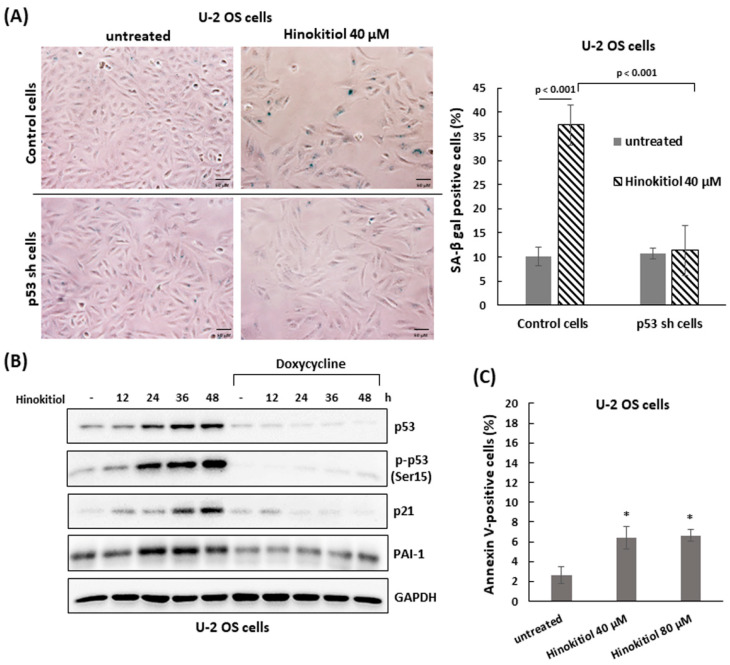
Hinokitiol triggers p53-dependent senescence in U-2 OS cells. (**A**) Hinokitiol enhances activity of senescence-associated β-galactosidase (SA-β-gal) in U-2 OS cells. Control and shp53 U-2 OS cells were incubated with hinokitiol (40 μM) for 48 h. Senescent cells were determined through SA-β-gal staining. Quantitative results of SA-β-gal-positive cells (blue color) are shown in the right panel. (**B**) Hinokitiol up-regulates p53 expression in U-2 OS cells. Control or shp53 U-2 OS cells were incubated with hinokitiol (40 μM) and collected at the indicated time points. Western blotting was used to analyze the level of p53, phospho-p53 (Ser15), PAI-1, and p21. (**C**) A small fraction of cells undergoes apoptosis in hinokitiol-treated U-2 OS cells. The cells were incubated with hinokitiol for 48 h. The number of apoptotic cells was determined by double staining of Annexin V and PI using flow cytometry. Quantitative data of Annexin V positive cells is revealed. * *p* < 0.001.

**Figure 5 ijms-23-01632-f005:**
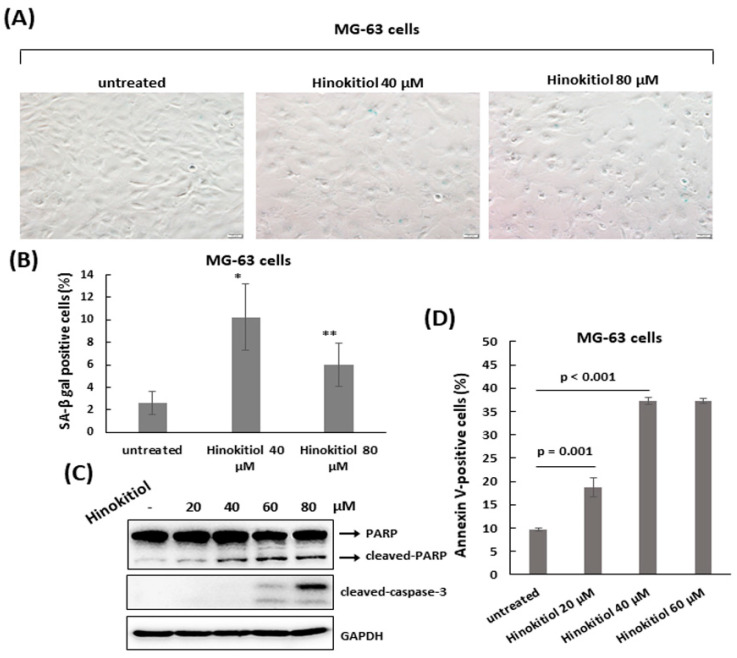
Hinokitiol induces apoptosis in MG-63 cells. (**A**) A small fraction of cells undergoes senescence in hinokitiol-treated MG-63 cells. MG-63 cells were incubated with different dose of hinokitiol for 48 h. Senescent cells were identified by SA-β-gal staining. Quantitative data of SA-β-gal-positive cells (blue color) was revealed (**B**). * *p* < 0.001, and ** *p* < 0.01. To detect apoptosis, MG-63 cells that were treated as in (**A**) were analyzed by Western blotting using the indicated antibodies (**C**), or stained with Annexin V and PI, and analyzed using flow cytometry. Quantitative data of Annexin V-positive cells are revealed (**D**).

**Figure 6 ijms-23-01632-f006:**
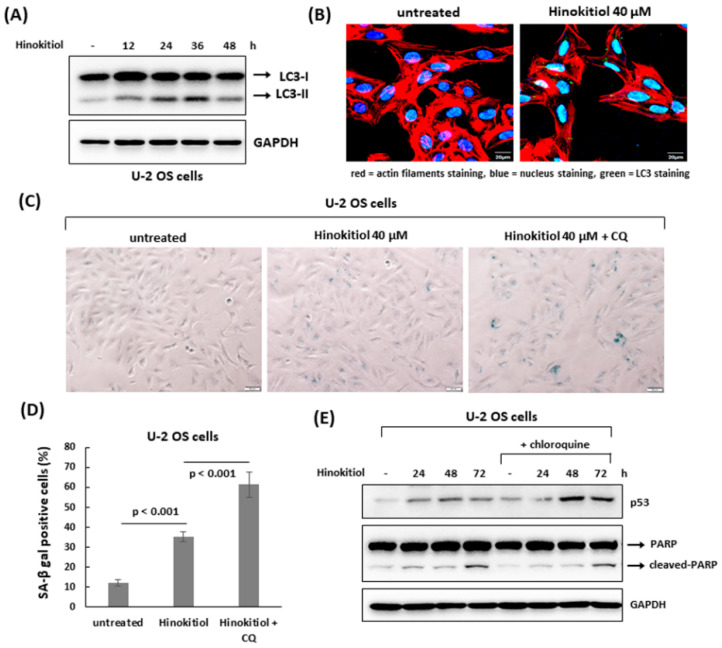
The suppression of autophagy enhances hinokitiol-induced senescence in U-2 OS cells. (**A**) Hinokitiol treatment triggers autophagy in U-2 OS cells. The cells were incubated with hinokitiol (40 μM), and harvested at the indicated time points. To detect autophagy formation, the level of LC3 protein was examined by Western blotting. (**B**) Hinokitiol treatment up-regulated autophagosome formation in U-2 OS cells. The cells were incubated with hinokitiol (40 μM) for 24 h. To measure LC3 puncta formation, the treated cells were immunostained with LC3 antibody for autophagy formation, phalloidin-iFluor 594 reagent for labeling actin filaments, and DAPI for labeling nuclei. (**C**) Treatment with chloroquine, an autophagy inhibitor, up-regulates hinokitiol-induced senescence in U-2 OS cells. The cells were incubated with hinokitiol (40 μM) for 48 h in the presence or absence of chloroquine (CQ) (5 μM). Senescent cells were identified by SA-β-gal staining. Quantitative data of SA-β-gal-positive cells (blue color) were revealed (**D**). (**E**) Administration of chloroquine up-regulates p53 expression in U-2 OS cells. U-2 OS cells were incubated with hinokitiol (40 μM) with or without chloroquine (5 μM). The cells were collected at the indicated time points and examined by Western blotting.

**Figure 7 ijms-23-01632-f007:**
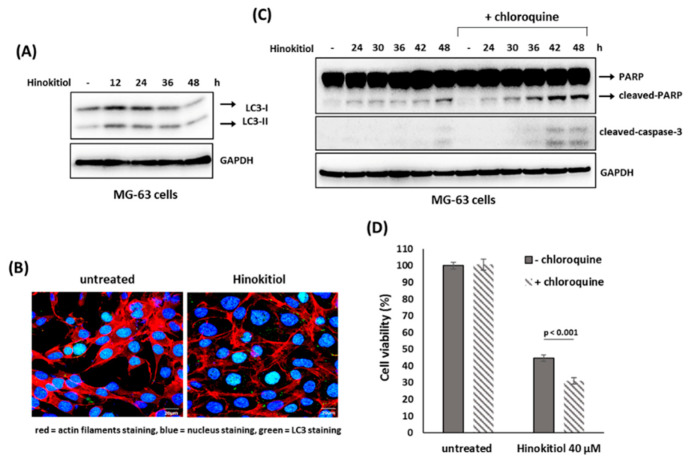
Suppression of autophagy enhances hinokitiol-induced apoptosis in MG-63 cells. (**A**) Hinokitiol treatment induces autophagy in MG-63 cells. To detect autophagy formation, MG-63 cells were treated and analyzed as described in Figure 6A. (**B**) Hinokitiol treatment up-regulated autophagosome formation in MG-63 cells. To measure LC3 puncta formation, MG-63 cells were treated and analyzed as described in Figure 6B. (**C**) Treatment with chloroquine up-regulates hinokitiol-induced apoptosis in MG-63 cells. The cells were incubated with hinokitiol (40 μM) with or without chloroquine (CQ) (5 μM) and collected at the indicated time points. The levels of cleaved caspase 3 and PARP were examined by Western blotting. (**D**) Administration of chloroquine decreases cell viability in hinokitiol-treated MG-63 cells. MG-63 cells were incubated hinokitiol (40 μM) for 48 h with or without chloroquine (5 μM). Cell viability of the treated cells was determined by MTT assay.

**Figure 8 ijms-23-01632-f008:**
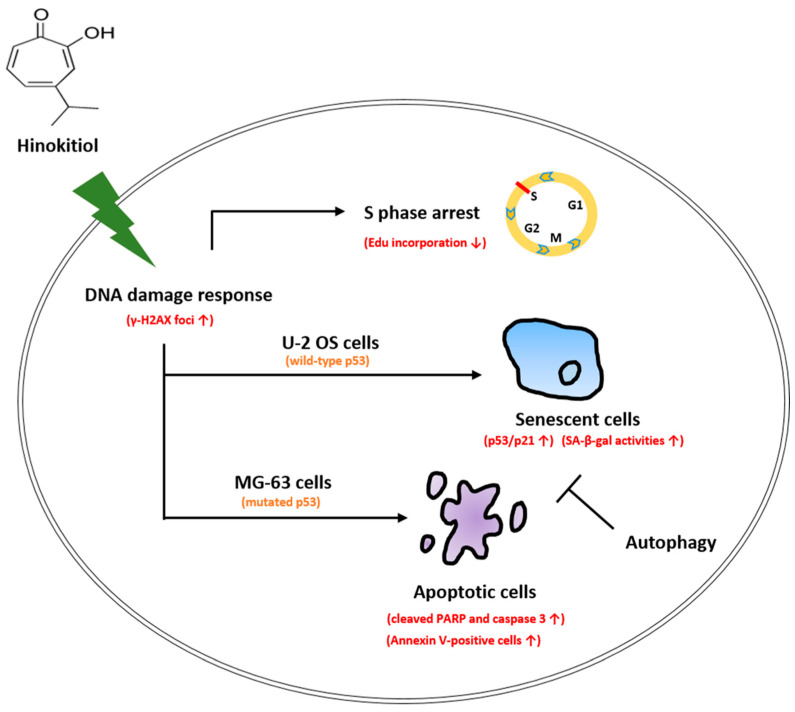
Schematic presentation of the hypothetical mechanisms for the role of hinokitiol in inhibiting osteosarcoma cell lines.

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
