# Peer review of "Different Cell Responses to Hinokitiol Treatment Result in Senescence or Apoptosis in Human Osteosarcoma Cell Lines"

_ijms, 2022, doi:10.3390/ijms23031632_

Round 1

Reviewer 1 Report

The aim of the study was to examine the cytotoxic activity of hinokitol against osteosarcoma (OS) cell lines with different p53 profiles and to illustrate molecular mechanisms of cell death in these cell lines. 

I have the following concerns and suggestions about this manuscript.

1) Figure 1A illustrated the structure of hinokitol. It is not novel, Since the authors did not perform structure-activity analysis, this figure looks pointless. 

2) Despite U-2 OS and MG-63 cells differing from each other in terms of p53 status, WB in Figure 2 shows that they respond in a similar way to hinokitol treatment (e.g. expression of pATM, Chk1, and 2 were increased). The explanation of this fact is missing.  

3) The image illustrating the strong increase of b-galactosidase expression in U-2 OS cells treated with hinokitol (Fig. 4A) is not convincing and has to be improved to be in a proper fit with text shown in the line 150.

4) Since beta-gal staining, shown in Figure 4A is not convincing, the authors also  have to show the expression of the other senescence-associated markers in hinokitol-treated OS cells (e.g. p16 lnkA, Lamin B1, etc.).

5) An increased expression of p53 is not a correct marker of p53 activation (as the authors mentioned in line 154). For this purpose, the authors have to examine the expression of phosphorylated forms of p53 (e.g. p53 Ser15) and include this data in the Figure 4B. 

6) The introduction is very weak and has to be expanded to illustrate the anti-tumor activities of hinokitol and the role of p53 in the regulation of apoptosis and senescence. 

7) Since the authors examined the expression of DNA damage repair (DDR) proteins in hinokitol-treated OS cells, they also have to address the issue of the type of DNA damage is being induced by hinokitol. For this purpose,  the authors have to run additional experiments. For example, single-gel electrophoresis (Comet assay) will be informative and helpful to delineate the possibilities between the formation of the single and double-stand breaks. Similarly, the expression of the other activated kinases (e.g. ATR and DNA-PK) will be also helpful.     

Minor:

1) The manuscript has some typos - e.g.  - line165 "p53-depnent"

2) Some of the figures do not have the reference in the text - e.g. Figure 4C

3) the illustrate the pro-apoptotic effects of hinokitol the authors are using Abs against cleaved PARP and caspase-3. Some of the figures contain the single marker (e.g. PARP - Figure 5C, 6E), the others contain both markers (e.g. Figure 7C). The authors have to make these figures uniform. 

Reviewer 2 Report

Overall summary

The information that is presented in this manuscript suggests that hinokitiol, a natural monoterpenoid isolated from heartwood of cupressaceous plants, has antitumor activities against human osteosarcoma cells (U-2 OS and MG-63 cells). Concretely, it was demonstrated its involvement in osteosarcoma cell growth and proliferation, at least by inducing cell-cycle arrest at S phase in a dose-dependent manner. Likewise, it has been shown that in U-2 OS cells hinokitiol exposure triggers cellular senescence, while it induces apoptosis in MG-63 cells. Furthermore, the underlying mechanisms of these effects were partly clarified.

General comments

The experiments support the conclusions that are made by the authors and the article is clear and interesting to read. However, there are a few concerns that should be addressed prior to this paper is considered for publication:

A) In relation to the section “Introduction”:

Lines 58-60. The sentence “Literature shows that hinokitiol suppresses cell viability and decreases survivin expression by ERK/MKP3/proteosome pathway in B16 melanoma” is not supported by the reference provided by Belayneh et al. (2021). It must be changed.

B) In relation to the section “Results”:

  1. The authors must indicate if the effect of hinokitiol has been investigated on normal human cells and if it is cytotoxic at concentrations used in this study.
  2. Lines 93-96. Why the colony formation assay was only performed with U-2 OS cells?
  3. Lines 97-98. The concentration of 40 µM was only chosen for the assays done with U-2 OS, wasn´t it?
  4. It is confusing to include information about MG-63 cells in the Figure 6E because the caption of the Figure 6.

C) In relation to the section “Materials and methods”:

  1. How many passages have you subjected your cell lines to? This information must be included in the subsection “cell culture and reagents”.
  2. The SA-β-gal staining must be explained. It is missing in the section “materials and methods”.

D) In relation to the section “Discussion”:

  1. Please, include a brief explanation about potential mechanisms that could explain different responses of tumor cells when the autophagy is inhibited.
  2. Since inflammation is a critical component of tumour progression, this relationship should be addressed in the discussion section, at least for the MG-63 cell line. Hinokitiol's antiinflammatory properties on this cell line have been studied previously: Shih et al. Evaluation physical characteristics and comparison antimicrobial and anti-inflammation potentials of dental root canal sealers containing hinokitiol in vitro. PLoS One. 2014;9(6):e94941. doi: 10.1371/journal.pone.0094941.

E) In relation to the section “References”: Please, review references #16 and #17 (abbreviated journal title).

Other comments:

  • Line 91: Change “was” by “were”
  • Line 69: Change “are” by “is”
  • Line 108. Delete one “.”
  • Figure 1. Change “U2-OS” by “U-2 OS”
  • Figure 4C was not referred to in the main body of the text
  • Lines 165 and 276. Change “depnent” or “depnedent” by “dependent”
  • Lines 164, 179, figures 6 (lines 209-222) and 7 (224-234). Change “µm” by “µM”
  • Line 205. Change “decrease” by “decreased”

Round 2

Reviewer 1 Report

The authors addressed properly to the comments and suggestions. The quality of the manuscript was improved.

The only issue that remains unresolved - is the type of DNA damage induced by hinokitol in cancer cells. It remains important since the authors provide evidence of activation of DDR, apoptosis, and senescence. The authors also agreed with this point and mentioned this in Discussion - lines 282-283.

Author Response

Comment: The only issue that remains unresolved - is the type of DNA damage induced by hinokitol in cancer cells. It remains important since the authors provide evidence of activation of DDR, apoptosis, and senescence. The authors also agreed with this point and mentioned this in Discussion - lines 282-283.

Answer: It is apologetic that we do not have ability to perform some experiments  which could directly analyze what type of DNA damage. In this report, the activation of ATM/Chk2 and ATR/Chk1 may provide the hints indirectly for this question.  Thanks for the understanding. 

Reviewer 2 Report

The manuscript has been improved and the authors have addressed the changes which were previously suggested. I have only one comment to make in relation to the following:

It is confusing to include information about MG-63 cells in the Figure 6E because the caption of the Figure 6.

Answer: About the information of MG-63 cells in Figure 6E, we want to explain this cell line is p53-null indeed. There is no paper to show p53 status in MG-63 cells by Western blotting.

In this case, the authors should change the caption of the Figure 6

Author Response

Comment: In this case, the authors should change the caption of the Figure 6

Answer:  Thanks for the comment. To avoid confusion, the information of MG-63 cells in Figure 6E was removed. The original caption of Figure 6 was maintained. 

Please see the Figure 6 in new version. 

This manuscript is a resubmission of an earlier submission. The following is a list of the peer review reports and author responses from that submission.

Round 1

Reviewer 1 Report

In this manuscript Shun-Cheng et all seek to evaluate the molecular basis of the anti-proliferative/anti-cancer activity of Hinokitiol. The experiments are straight-forward, and presentation is relatively easy to follow although some English editing will be needed.

There are two major issues (both fatal) with this paper.

  • The activity of this agents against cancer cell lines in this study is mid-double digit micromolar range. At this level of activity Hinokitol has no value as a bona-fide anti-cancer agent.
  • Hinokitiol, is a simple molecule that is unlikely to present sufficient binding energy to any specific target or a family of targets. Therefore, we can not use Hinokitol as a probe to study potential approaches to cancer treatment either.

These two major flaws dramatically reduce the scientific value of this paper for community.

There are also a few minor issues that reviewers should pay attention to.

  • Figure 1B contains data for both U2-OS and MG-63 cells. Figure 1C only U2-OS cells; it is not a logical experimental set-up.
  • S-phase arrest (depending on the mechanism of arrest) can cause DNA damage and thereby activate DNA damage check point, not the other way around.
  • Both cells (p53 mutant and WT) show very similar DNA damage checkpoint response. Is that response really related to p53? Perhaps this issue should have been further investigated using U2-OS cells with or without knocking down p53.
  • What is the importance of senescence in the anti-cancer activity of hinokitiol? While knocking down p53 will inhibit senescence it is not at all clear that it contributes to anti-cancer activity of this agent, at least in this model.
  • Figure 5 o9nly employs MG-63 cells. Why are U2-OS cells excluded from these experiments?
  • Figure 6 and 7 deals with hypothesis that suppression of autophagy will improve induction of senescence or apoptosis in U2-OS and MG-63 cells, respectively. Are the effects additive or synergistic? This interesting study is preliminary and in deed of significant refinement.

Reviewer 2 Report

This study shows the effect of hinokitiol on osteosarcoma cell lines. The conclusion of this study is that hinokitiol may provide a novel treatment strategy for osteosarcoma. A major weakness of this study is that alll expermiments are in vitro experiments performed on osteosarcoma cell lines. However, since this is a new compound tested on osteosarcoma cell line it is worthwhile publishing the data as is. the authors, however, should add a paragraph mentioning the limitation of this study.

Author Response

Comment: A major weakness of this study is that alll expermiments are in vitro experiments performed on osteosarcoma cell lines. However, since this is a new compound tested on osteosarcoma cell line it is worthwhile publishing the data as is. the authors, however, should add a paragraph mentioning the limitation of this study.

Answers: Thanks for the suggestion. The sentences were added in third paragraph of Discussion part that described as follow:

“However, the limitation of this study is that all results were from in vitro experiments. Recent literatures have shown anti-cancer activity of hinokitiol against different tumors in mouse model. In these studies, the effective dose was ranged from 2 mg to 100 mg/kg/day in mice, and the data shows that hinokitiol is relatively nontoxic to the animals even in higher dose. Whether hinokitiol could inhibit the growth of osteosarcoma and what is effective in vivo dose remains to be demonstrated.”

Round 2

Reviewer 1 Report

The authors responses do not address the issues I have raised. 

I am quite perplexed by the authors responses. Are the authors arguing that Hinokitiol is a potential anti-cancer agent for treatment of bone cancer because these authors demonstrated 10 years ago thatit has  5-10 uM IC50 in colon cancer cells and therefore the current manuscript should be accepted? If I were to accept this rather unusual argument then what is the point of current manuscript? Are authors expecting the conduct this very same study in 100 different types of cancers and publish 100 papers regardless what the effective doses are? There is a reason we treat hormone receptor positive and negative cancers of the same organ with different drugs. Every cancer is different, every sub-type of cancer is different.

The related argument that in vivo effective dose of this agent will be lower than its effective dose in vivo is not based on any data but wishful thinking.

Similarly just because Tropotone had certain activity does not mean Hinokitoil will have the same. These two molecules have very different electronics. Tropotone can form metal complexes that may or may not be formed by Hinokitiol, even if they are formed the electronics of resulting complexes and their physical shapes will be very different. This will be reflected in their compatibility for and binding energies to specific pockets of a potential macromolecular target.